# The feasibility, appropriateness, and usability of mobile neuro clinics in addressing the neurosurgical and neurological demand in Uganda

Benjamin Mukumbya[1,2‡], David Kitya[1,3,4‡], Yesel Trillo-Ordonez[1,2], Keying Sun[2,5], Oscar Obiga[1,6], Di D. Deng[1], Kearsley A. Stewart[2], Alvan-Emeka K. Ukachukwu[1,2,7]*, Michael M. Haglund[1,2,7], Anthony T. Fuller[1]

1 Duke Global Neurosurgery and Neurology, Durham, NC, United States of America, 2 Duke Global Health Institute, Durham, NC, United States of America, 3 Department of Neurosurgery, Mbarara Regional Referral Hospital, Mbarara, Uganda, 4 Mbarara University of Science and Technology, Mbarara, Uganda, 5 Duke University School of Medicine, Durham, NC, United States of America, 6 Department of Neurosurgery, Mulago National Referral Hospital, Kampala, Uganda, 7 Department of Neurosurgery, Duke University Health System, Durham, NC, United States of America

‡ BM and DK are co-first authors on this work.
* alvan.ukachukwu@duke.edu

## Abstract

### Introduction

Uganda has a high demand for neurosurgical and neurological care. 78% of the over 50 million population reside in rural and remote communities where access to neurosurgical and neurological services is lacking. This study aimed to determine the feasibility, appropriateness, and usability of mobile neuro clinics (MNCs) in providing neurological care to rural and remote Ugandan populations.

### Methods

Neurosurgery, neurology, and mobile health clinic providers participated in an education and interview session to assess the feasibility, appropriateness, and usability of the MNC intervention. A qualitative analysis of the interview responses using the constructs in the updated Consolidated Framework for Implementation Research was performed. Providers' opinions were weighted using average sentiment scores on a novel sentiment-weighted scale adapted from the CFIR. A stakeholder analysis was also performed to assess the power and interest of the actors described by the participants.

### Results

Twenty-one healthcare providers completed the study. Participants discussed the potential benefits and concerns of MNCs as well as potential barriers and critical incidents that could jeopardize the intervention. Of the five CFIR domains evaluated, variables in the implementation process domain showed the highest average sentiment scores, followed by the

**Data Availability Statement:** All relevant data are included within the manuscript and its supporting

information files. In addition, relevant deidentified data files containing all coded interviews, domain analysis, and data analysis codes are available from the Duke University Research Data Repository, and are accessible via the DOI: https://doi.org/10.7924/r4m90f080 (plain text: 10.7924/r4m90f080).

**Funding:** BM received funding from the Duke Global Health Institute (DGHI), [website: [https://globalhealth.duke.edu].

**Competing interests:** The authors have declared that no competing interests exist.

implementation climate constructs, inner setting, innovation, and outer setting domains. Furthermore, many interested stakeholders were identified with diverse roles and responsibilities for implementing MNCs. These findings demonstrate that MNC innovation is feasible, appropriate, and usable.

## Conclusion

The findings of this study support the feasibility, appropriateness, and usability of MNCs in Uganda. However, integration of this innovation requires careful planning and stakeholder engagement at all levels to ensure the best possible outcomes.

## Introduction

Neurological disorders are among the leading causes of death and disability worldwide, with over 22 million people living with these disorders and 5 million deaths attributed to them annually [1, 2]. Access to neurological (synonymous with neurosurgical and neurological) care is disproportionately low in rural and remote settings, where healthcare barriers are prominent. Uganda is a low-income country (LIC), home to more than 50 million people and 1.5 million refugees [3, 4]. Furthermore, the World Poverty Clock (2023) estimates that 16% of Uganda's population is impoverished, and 78% live in rural and remote communities where access to essential health care is limited or non-existent due to a variety of existing barriers [5, 6]. Given that Uganda only has four neurological hospitals, providing neurological care across the country is impossible [7–12]. Uganda's demand for specialized neurological services exceeds its supply due to inadequate infrastructure and workforce [13, 14]. The aforementioned barriers lead to disparities in the quality of neurological care for people in rural and remote communities, signifying an urgent need for innovative approaches to neurological service delivery. These approaches should transcend geographical isolation and socioeconomic factors to address challenges such as limited access to neurological care, shortage of trained healthcare professionals, and inadequate infrastructure.

Mobile health clinics (MHCs) provide healthcare services in remote communities using customized vehicles or mobile units [15]. They are staffed by healthcare professionals who offer a variety of medical services and are used in many settings globally to meet the demand for healthcare, especially in resource-limited settings [16–18]. MHCs have been used to bring essential medical services to underserved populations worldwide, including in the United States (US) [16, 17, 19] and Uganda, where Jatho and colleagues used MHCs to provide cancer management to patients [20]. MHCs can significantly improve patient outcomes by providing patient education, early screening, identification, follow-up, and rehabilitation services [16, 17, 20–23]. Qureshi and colleagues successfully implemented mobile neuroendoscopy in East Africa, which treated hydrocephalus patients and trained a neurological workforce [24, 25]. Therefore, implementing well-designed MNCs can enhance healthcare equity by eliminating barriers that hinder rural and remote communities from accessing neurological care [26].

Given the evident neurological burden, constrained resources, and the potential effectiveness of MHCs in addressing Uganda's challenges, the adaptation of MNCs becomes imperative [27, 28]. A pediatric neurosurgical MHC successfully designed and implemented by Owler and colleagues in rural and remote Australia improved neurosurgical outcomes for patients by decreasing the delays to definitive neurological care [21]. Some facilities in Uganda, such as CURE Children's Hospital of Uganda (CCHU), have integrated mobile clinic models into

neurosurgery practices, incorporating tools such as mobile endoscopy [25, 29–32]. The purpose of this study was to comprehensively analyze the potential and practicality of MNCs as a strategic intervention for improving neurological care, particularly in Uganda's constrained remote and rural areas.

## Methods

Study participants were interviewed after attending an educational session introducing the Mobile Neuro Clinics (MNCs) concept. The Consolidated Framework for Implementation Research (CFIR) domains and background literature from MHCs, neurosurgery, and neurology were used to design the educational content and interview guides [33–37]. Our approach aimed to align the study with a strong theoretical framework while ensuring the relevance and contextual applicability of the educational content (https://youtu.be/55fRRXSyMks) and (S1 File). The CFIR framework comprehensively evaluates innovations and their implementation in specific contexts [38–40]. As such the CFIR framework enables a holistic examination of the complex interplay of factors influencing MNCs and their successful implementation. Furthermore, the study included healthcare providers with varying experience levels, the CFIR framework was ideal for thoroughly evaluating the participants' perspectives. By applying the CFIR guidelines, the study identified potential barriers and facilitators to innovation and further assessed Uganda's current neurological services. This evaluation further identified whether MNCs are feasible, appropriate, and usable (S1 Table).

### Setting and participants

The study was conducted at Mulago National Referral Hospital (MNRH), Mbarara Regional Referral Hospital (MRRH), and other locations chosen by MHC providers in Uganda. MNRH is Uganda's largest public hospital, located in Kampala, with established neurological departments, and is affiliated with Makerere University. MRRH is a teaching hospital in western Uganda offering clinical services in mental health care, neurology, and neurosurgery. Mobile healthcare providers who were unable or preferred not to participate in the study at MNRH or MRRH were interviewed at a location of their choice.

Between July 27, 2022, and October 5, 2022, neurological specialists (neurosurgeons, neurologists, and residents) involved in patient care at MNRH and MRRH were recruited in the study. MHC administrators and healthcare professionals with at least one year of experience were also enrolled, within the same period, due to their understanding of how mobile health services operate in the country. All study participants were required to be competent in English. Participants who did not meet the above criteria were excluded.

Purposive sampling was used to select neurological and MHC providers for the study. Healthcare providers were contacted via phone, email, and in-person interactions and invited to attend an education session. Participants who attended the in-person education sessions at MNRH and MRRH were contacted to schedule interviews. Other participants who watched the YouTube education session (https://youtu.be/55fRRXSyMks) were also contacted for follow-up interviews. Prior research indicates that 12–16 interviews may provide at least 90% saturation; [41–43] however, the expected sample size for the recruitment pool of this study was 24 individuals.

### Ethics statement

The study received ethical approval from the Duke Health Institutional Review Board (IRB) (Pro00110750) and from the Mulago Hospital Research Ethics Committee (MHREC 2317). Both verbal and written informed consent were obtained from study participants and

documented on a pre-specified consent form approved by the two institutional ethical review boards. Study participants were compensated $35–52 USD upon their completion of the education session and interviews.

## Data collection

Data were collected from consenting participants using secure recording devices provided by Duke Global Neurosurgery and Neurology (DGNN). All interviews were conducted in English by a single researcher (B.M.) to ensure consistency. The interview audio was securely stored on a secure Duke Box cloud, with access restricted to approved team members for transcription. Three team members (B.M., Y.T.O., and K.S.) transcribed the recordings independently. The transcripts were then thoroughly verified for accuracy by the investigator (B.M.) who conducted the interviews in the field, and any important revisions were performed to ensure high-quality transcriptions.

We conducted in-depth face-to-face interviews in which participants answered open-ended questions before being probed to clarify any additional information. Interview guides collected demographic information, and also had four different sections. The four sections collected information related to the understanding of healthcare, MHCs and their application in neurological care and other general medical specialties, and the barriers and facilitators to designing and implementing MNCs. The first part of the interview guide for MHC providers assessed their general understanding of neurosurgery and neurology, while that of neurological specialists assessed their general understanding of mobile clinics. The remaining three parts of both interview guides followed the same structure.

## Data coding and analysis

We created two distinct codebooks from 5 sampled neurological and 4 sampled MHC providers' interviews, respectively. These two codebooks were subsequently combined to form a single comprehensive codebook (S2 File and S2 Table). We used applied thematic analysis to build these codebooks, integrating inductive and deductive reasoning [43–45]. We guided our deductive approach with the CFIR-assigned constructs while allowing the inductive approach to help identify any new and emergent themes that the CFIR constructs had not previously considered [31, 32, 44, 45]. After B.M. developed the codes, the group provided feedback on what needed to be improved. A team of four individuals (B.M., Y.T.O., K.S., and A.T.F.) utilized Microsoft Excel to formulate and refine the codes and themes for analysis. The team thoroughly examined the codes, their meanings, and how they related to the study's aims. After completing the full codebook, three team members (B.M., Y.T.O., and K.S.) individually coded the transcripts that were assigned to them. During team meetings, all four members reviewed the coded content to ensure the codes were correctly applied using the comprehensive codebook shown in **Fig 1** and **S2 File**.

## Sentiment weighted scale

Once the interviews had been transcribed, our team analyzed the data using a modified version of the sentiment-weighted scale (SWS) based on CFIR [33, 34, 46]. We adopted the SWS to assign sentiment ratings (SR) to the codes and themes that emerged from the interviews, considering the magnitude of each construct and its valence (positive or negative influence), [33, 34, 47] as shown in **Table 1**.

To evaluate the MNC concept's feasibility, appropriateness, and usability, we compared log-transformed average sentiment scores (ASS) across five CFIR areas. Analyzed CFIR domains include the innovation domain, implementation climate construct, outer setting

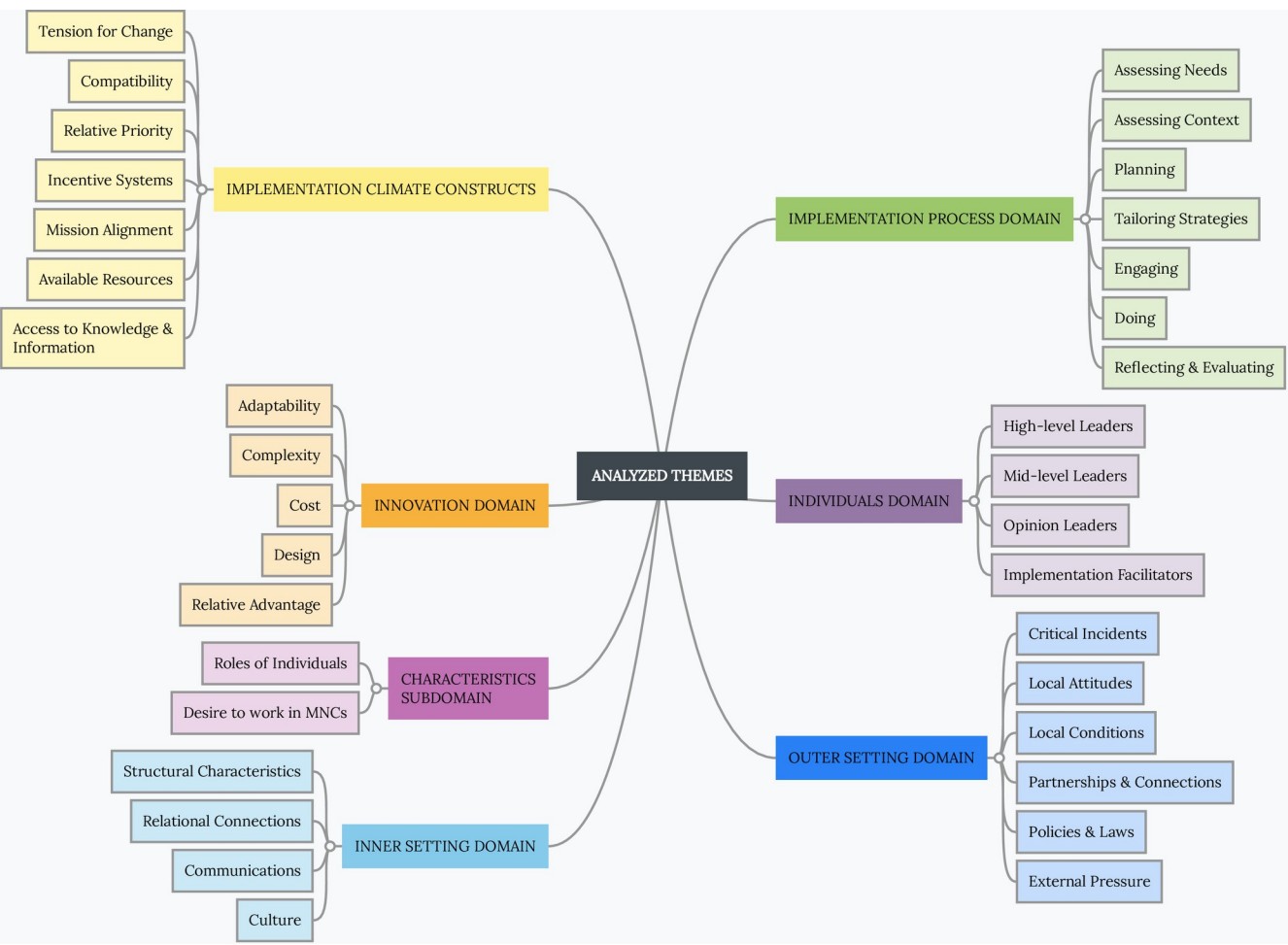

**Fig 1. Summary of analyzed themes.** A detailed description of the themes, codes, and sub-child codes is in S1 and S2 Tables. The ***Innovation Domain*** describes all the characteristics of Mobile Neuro Clinics or Mobile Health Clinics and then goes on to explain how the intervention can be used to meet the demand for local neuro care. The ***Outer Setting Domain*** is the setting in which the ***Inner Setting*** exists: Uganda, international communities, and any other Communities. The ***Inner Setting Domain*** is the setting in which the innovation (MNCs) is intended to be implemented: rural and remote communities in Uganda. ***Implementation Climate Constructs*** characterize the ability to absorb change, the providers' shared openness to MNCs, and the degree to which MNC use will be rewarded, encouraged, and anticipated. ***Individuals Domain*** describes the roles and characteristics of individuals. The ***Characteristics subdomain*** describes the MNC characteristics applicable to the roles in the MNC project. The ***Implementation Process Domain*** explains the procedures and strategies used to put the innovation into practice. The domain recorded the activities and strategies that would be used to implement the innovation, as well as the implementation process framework.

domain, inner setting domain, and implementation process domain. The ASS was defined as the average SR for the participants that responded to that theme or code. Computation and visualization were performed in R (*The R Foundation for Statistical Computing, Vienna, Austria*).

## Stakeholder analysis

The team further determined the identified stakeholders' power, interest, and roles based on the CFIR guidelines for the Individuals domain and the Characteristics subdomain [47–49]. Interviews with participants led to the identification of four major stakeholder types. High-level Leaders were defined as important decision-makers, executive leaders, directors, or government figures who have the power to influence the adoption of innovations [33, 34, 46–50].

**Table 1. Sentiment weighted scale.**

| | Sentiment Weighted scale modified for the study from CFIR Ratings | | | | |
|---|---|---|---|---|---|
| **weighted scale** | highly unfavorable | unfavorable | neutral | favorable | highly favorable |
| **numeric sentiment ratings** | -2 | -1 | 0 | 1 | 2 |
| **definitions** | Very highly opposing views to designing & implementing MNCs | Opposing views to designing & implementing MNCs | No data reported | Supporting views to designing & implementing MNCs | Very highly supporting views to designing & implementing MNCs |
| **criteria** | Detailed evidence with examples | Evidence with at least an example | .. | Evidence with at least an example | Detailed evidence with examples |
| **Evaluation for Feasibility, Appropriateness & Usability** | | | | | |
| **Term** | **Equation** | | | **Note** | |
| **Average Sentiment Scores (ASS)** | $ASS = \dfrac{\Sigma(sentimen\ ratings,\ sr)}{Total\ respondents}$ | | | Total respondents = number of participants who responded to the theme/ codes in that domain. | |

**If the overall positive ASS was greater than the negative ASS for a given CFIR construct /domain, that CFIR construct/domain was considered feasible, appropriate, and as such, the innovation, MNC, was considered usable.

Mid-level Leaders were managers responsible for supervising subordinates and were managed by high-level leaders [33, 34, 46–50]. Opinion Leaders describe stakeholders as those who exert subtle influence on the actions and thoughts of others [33, 34, 46–50]. Leaders in this category include religious leaders, local leaders, and traditional and cultural leaders. Implementation Facilitators described field experts who would provide guidance, coaching, or support during the implementation process [33, 34, 46–50]. This category included community health workers (CHWs), village health teams (VHTs), and medical officers.

The combination of sentiment and stakeholder analysis provides a comprehensive understanding of the challenges and opportunities faced by innovations while identifying key players that are critical to their successful establishment and operations.

### Inclusivity in global research

Additional information regarding the ethical, cultural, and scientific considerations specific to inclusivity in global research is included in the Supporting Information (**S3 File**).

## Results

### Demographics

A total of 21 healthcare professionals were interviewed. Twenty in-depth face-to-face interviews ranging from 20 minutes to 1 hour and 30 minutes were conducted, excluding the educational session components. Nineteen (95%) of the 20 interviews comprised distinct, in-depth face-to-face interviews, with two participants (5%) interviewed in the same session. Seven (33.3%) MHC providers and 14 (66.7%) neurological providers were interviewed.

The participants were between the ages of 25 and 65, with a mean age of 37.9 years and a median age of 35 years. Only three participants (14.3%) were female. Ten (47.6%) of the participants had only neurological care experience; eight (38.1%) had both neurological and MHC experience; and three (14.3%) had only MHC experience. Additionally, two neurological specialists and six MHC providers had administrative authority. The time in the profession ranged from 8 months to 25 years, with a mean of 6·3 years and a median of 5 years (**Table 2**).

**Table 2. Demographics and experience of participants.**

| Demographics | | | |
|---|---|---|---|
| | **Men** | **Women** | **All** |
| **Age (years)** | N (%) | N (%) | N (%) |
| **25–35** | 7 (33.3) | .. | 7 (33.3) |
| **35–45** | 8 (38.1) | 2 (9.5) | 10 (47.6) |
| **45–55** | 1 (4.8) | .. | 1 (4.8) |
| **55–65** | 2 (9.5) | 1 (4.8) | 3 (14.3) |
| **All Participants** | 18 (85.7) | 3 (14.3) | 21 (100) |
| **Time Served by Providers (years)** | | | |
| | **MHC Providers** | **Neurological Providers** | **Total** |
| Time Served Interval (years) | N (%) | N (%) | N (%) |
| **0.0–5.0** | 4 (19.0) | 6 (28.6) | 10 (47.6) |
| **5.0–10.0** | .. | 6 (28.6) | 6 (28.6) |
| **10.0–15.0** | 2 (9·5) | 2 (9.5) | 4 (19.0) |
| **25.0–30.0** | 1 (4.8) | .. | 1 (4.8) |
| **Total** | 7 (33.3) | 14 (66.7) | 21 (100) |
| **Experience of Providers** | | | |
| **Age Distribution** | | | |
| Range (years) | | | 25–63 |
| Mean (years) | | | 37.9 |
| Median (years) | | | 35.5 |
| **Providers by Profession** | | | |
| MHC Providers (N, %) | | | 3 (14.3) |
| Neurological Providers (N, %) | | | 10 (47.6) |
| Both MHC and Neurological providers (N, %) | | | 8 (38.1) |
| **Time in Profession** | | | |
| Range | | | 8 months—25 years |
| Mean (years) | | | 6.3 |
| Median (years) | | | 5 |

## Sentiment weighted scale results

We evaluated the sentiment ratings for five CFIR domains based on healthcare providers' perceptions in our study. The implementation process domain variables had the highest ASS. The implementation climate constructs, inner setting domain, outer setting domain, and innovation domain ASSs followed sequentially. The perspectives of healthcare practitioners across all CFIR domains had a stronger positive valence, as observed in the boxplot in (**Fig 2A**). The positive valence of the SWS describes the significance of MNCs in terms of feasibility, appropriateness, and usability.

## Innovation domain

Innovation domain findings identified substantial insights into the fundamental components of MNCs from the perspective of providers. Six themes/codes (54.5%) scored high positive valency, while five (45.5%) scored negative valency (**Fig 2B**). Opinions from both providers show that there are numerous perceived benefits to the innovation (**Fig 3**). Additionally, ASS findings suggest that the providers support the idea that the innovation can be designed with numerous central, adaptable components in an appropriate amount of time to address

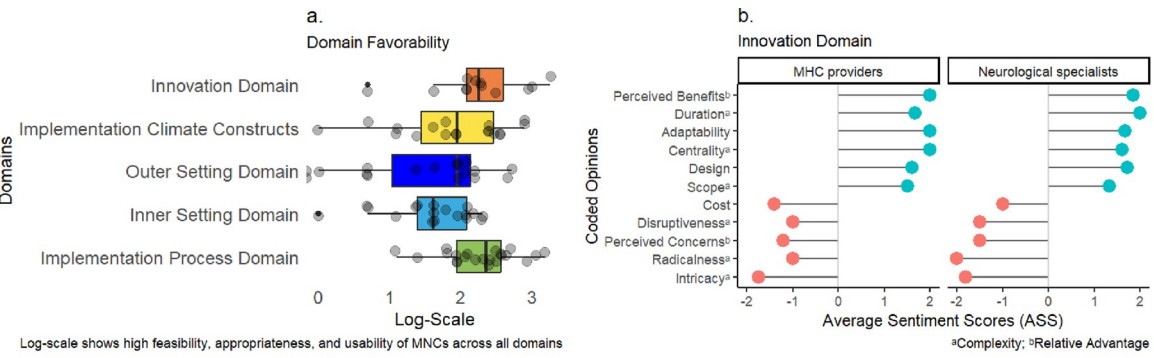

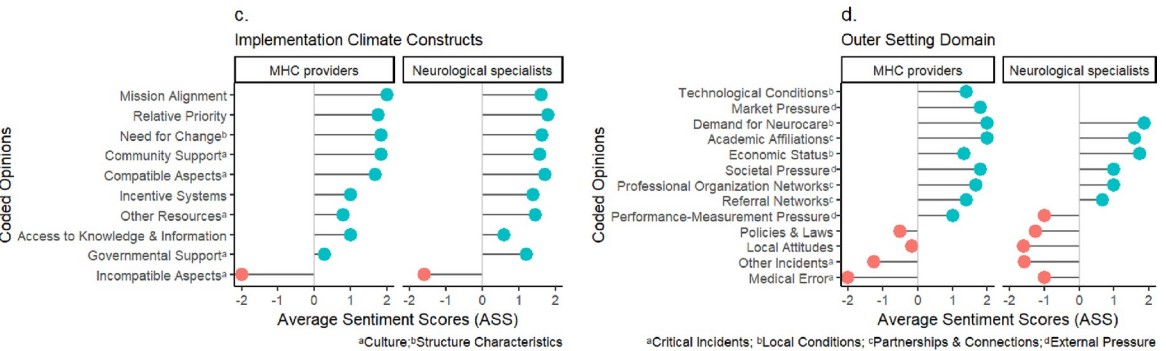

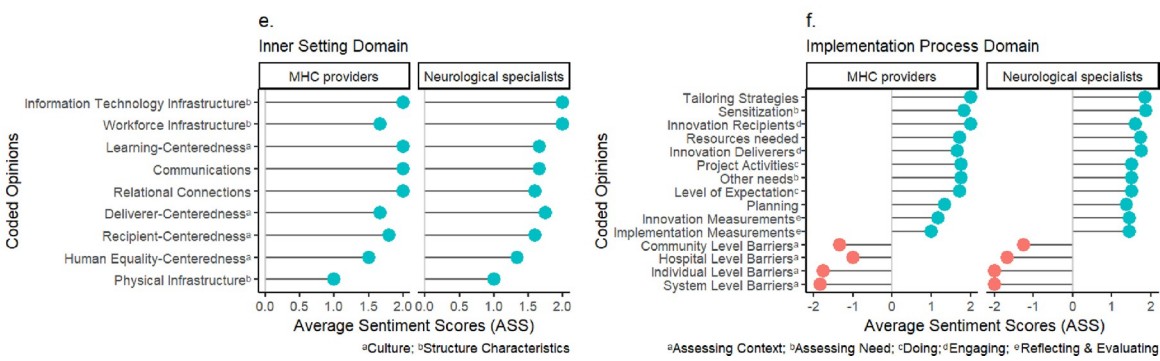

**Fig 2. CFIR sentiment-analyzed domains. Fig 2A**, the *Domain Favorability*, succinctly encapsulates providers' enthusiastic embrace of innovative MNCs. The findings indicate a significant support for the MNC idea in the analyzed setting. **Fig 2B**, the *Innovation Domain*, delineates critical features of concern and facilitators pertaining to both MNCs and MHCs. Providers further elucidate how the intervention would address local neurological care demand. The *Implementation Climate Constructs* in **Fig 2C** demonstrate the high flexibility to change, providers' collective receptivity to MNCs, and the level to which MNC adoption would be acknowledged, promoted, and anticipated. **Fig 2D**, the *Outer Setting Domain*, depicts Uganda's, foreign communities', and other locations' roles in assisting MNCs. While impediments exist, various facilitators were identified that could support MNCs. In the *Inner Setting Domain* (**Fig 2E**), the focus was on the specific implementation setting, namely rural and remote communities in Uganda. Providers exhibit an exceptionally optimistic view in this domain regarding the MNC concept. Finally, **Fig 2F**, the *Implementation Process Domain*, delineates the procedural and strategic aspects of introducing the innovation. Providers discussed key activities and strategies essential for successful implementation, alongside an overarching implementation process framework.

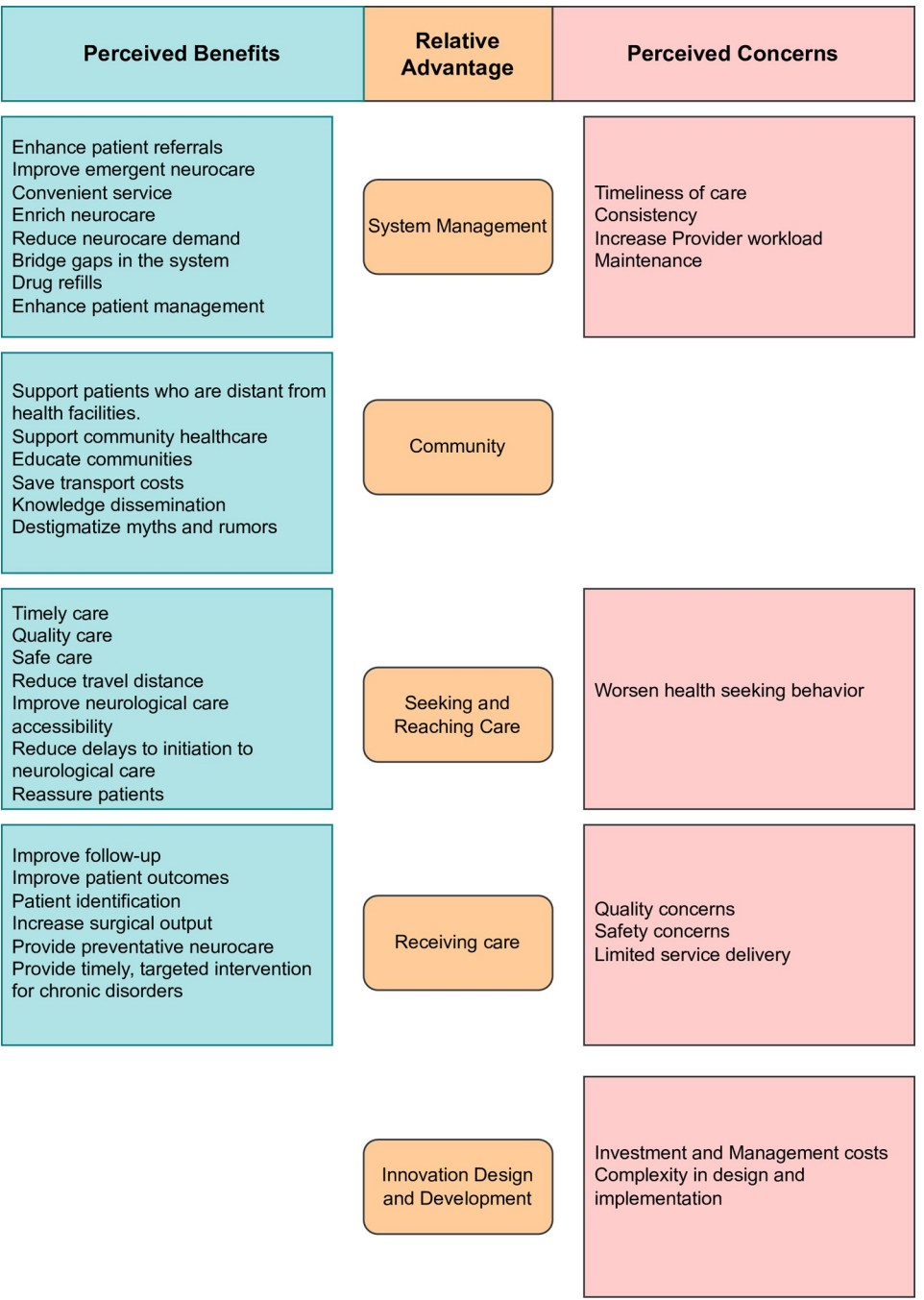

**Fig 3. Relative advantages of mobile neuro clinics.** Fig 3 unveils the relative advantages associated with MNC innovation as described by healthcare providers. The findings highlight the positive aspects of MNCs, demonstrating how the perceived benefits far surpass any perceived concerns. This illustrative diagram meticulously delineates the key perceived benefits that can be attained through carefully designing and implementing MNCs.

neurological demands. The ASS, however, shows that the providers have numerous concerns, including the MNCs' cost, disruptiveness, radicality, and intricacy. **Fig 3** classifies the aspects of the relative advantage theme, including perceived benefits and concerns. Considering all factors, providers understand how critical it is to modify and rethink current mobile clinic

models for neurological treatment to create superior MNCs. With thoughtful designs and implementation strategies, these MNCs have the potential to significantly improve access to critical neurological care and meet the current demand for neurological services. A neurologist emphasized that:

> *"Mobile neuro clinics might not be as good as being in a hospital. However, they may go a long way toward alleviating some of the community's common conditions. Because [they] can treat and, more importantly, recognize the conditions. . . Then, because [they] identify cases and refer them to the center, [they] can undoubtedly improve referrals. In some ways, [they] also assist the staff and health care providers in rural areas. You also educate them on each condition. Teach them how to treat these conditions. As a result, it will greatly benefit the community." (Neurologist).*

### Implementation climate constructs

The implementation climate constructs explain how receptive healthcare providers are towards MNCs and how such innovation would be rewarded, supported, and expected. The research examined the implementation climate themes/codes and how participants anticipated how Ugandan communities and the government would respond to the intervention. Nine themes/codes (90%) scored a high positive valence, while only non-compatible aspects (10%) scored a negative valence. (**Fig 2C**). Respondents recognized the potential of MNCs as a two-pronged approach to alleviating shortages in neurological services. By delivering essential care directly to communities and emphasizing preventive measures, MNCs could benefit Ugandans at all levels of society. Nevertheless, providers expressed concerns about limited access to neuroscience knowledge, insufficient government support, and potential incompatibilities in the components essential for developing and implementing MNCs. **Fig 4** summarizes the variables in the compatibility theme. One neurosurgery resident explained that:

> *"Outside of Kampala, you will only see Mbarara and CURE in Mbale, and that's it. . . As a result, there is a significant deficit. . ." (Neurosurgery Resident).*

### Outer setting domain

Healthcare providers assessed Uganda's current condition and described potential unforeseen incidents related to the efficient implementation of MNC healthcare practices within the outer setting domain. **Fig 2D** shows nine (69.2%) themes/codes scored highly on the positive valence SWS for MHC providers and six (46.1%) for neurological specialists. Four (30.8%) themes/codes scored negatively on the valence SWS. The participants reinforced the need to recognize that, despite commendable efforts, the present neurological care initiatives are not up to standard with the expanding demands. Numerous determinants of neurological disorders are on the rise, necessitating continuous collaboration, innovation, and resource allocation to address the evolving healthcare needs of the Ugandan population. Most of the themes/codes in the outer setting domain support the successful implementation of the innovation concept, while the others describe aspects to be cautious about. A neurosurgeon explained that:

> *"We must ascertain how we fall under the legal purview of medical practice and which laws apply to mobile clinics. You are aware that any clinic operating within the nation is subject to some type of legal regulation. . . Because you need a license to register as a clinic right now.*

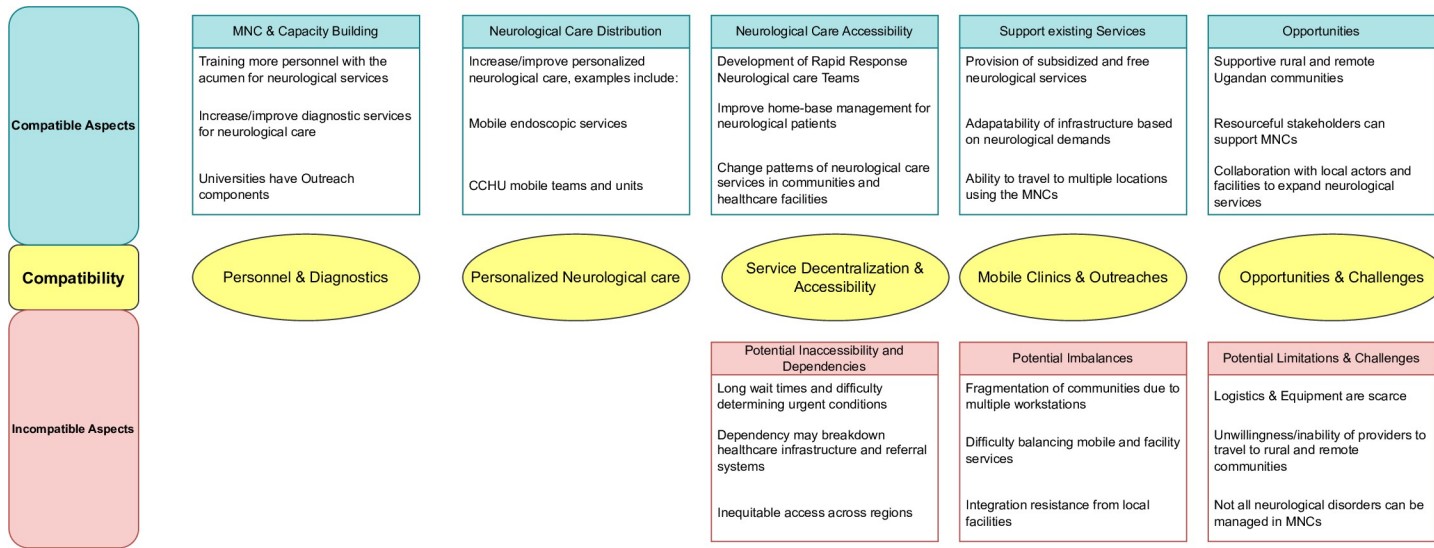

**Fig 4. Navigating the compatibility of mobile neuro clinics.** Fig 4 presents a nuanced examination of the compatibility of mobile neuro clinics based on insights shared by healthcare providers. The research findings accentuate a comprehensive and notable summary, revealing that the compatible aspects of MNCs significantly outweigh the incompatible aspects. This visual representation offers a valuable perspective for actors, suggesting that Uganda possesses numerous quantifiable aspects that can be utilized during the integration of MNCs.

*These licenses are now linked to URSB registration, which is one of the rules. You must register, and your clinic must be registered as a company with an address, so those are some of the problems you'll encounter. What's going to be the address of the mobile clinic?"* (Neurosurgeon).

## Inner setting domain

The inner setting domain examined healthcare providers' perspectives on Uganda's rural and remote communities. The domain evaluated providers' opinions on local communities and how they considered innovation could be beneficial to these regions. All the themes and codes scored highly in the positive valence on the SWS. Based on the ASS scores, the providers' opinions indicate that MNCs are culturally appropriate, and as such, strategies to allow MNCs to operate can be developed, as shown in **Fig 2E**. As a result, the providers emphasized the collaboration of MNCs and health centers.

*"Collaboration with the health system, I believe, is an important factor. The current structure of the health-care system includes either a health center III or a health center IV. That could be useful. It would be extremely beneficial. And then train the health center I in the village; there are village health teams (VHTs) to try to identify those other people in the community who need help but are unwilling to go to the hospital or change their attitude. As a result, VHTs are being incorporated. It would be ideal." (Neurologist).*

## Implementation process domain

The implementation process domain findings described potential barriers, efficient strategies, and the resources required for successful MNC implementation. Eleven (73.3%) themes or codes scored on the positive valence of the SWS, and only four (26.7%) scored on the negative

valence. (**Fig 2F**). According to the ASS, interviewees expressed numerous aspects that can be utilized to overcome the existing barriers. **Fig 5** depicts the barriers that might be encountered in the Ugandan context when MNC innovation is designed and implemented, while **Fig 6** summarizes the planning and strategies the providers suggested to utilize the limited resources to implement MNCs efficiently. Providers also recognize that part of the MNC intervention would involve community outreach and campaigns, among other activities.

> *"So those people who have an influence on the population, the local citizens, have to be really educated, so there are going to be massive community awareness exercises and campaigns for long, protracted times to get these people to understand the neurosurgical problems that need intervention." (Neurosurgeon).*

## Stakeholder analysis from individuals domain and sub-characteristics

The stakeholder analysis interpretations integrate perspectives based on individuals and sub-characteristic domains to examine the power and interests of key actors critical to the innovation. **Fig 7** presents numerous actors along with their respective roles and responsibilities essential for the implementation of MNCs in Uganda's communities. Providers encourage engaging stakeholders from the start, which is of prime importance.

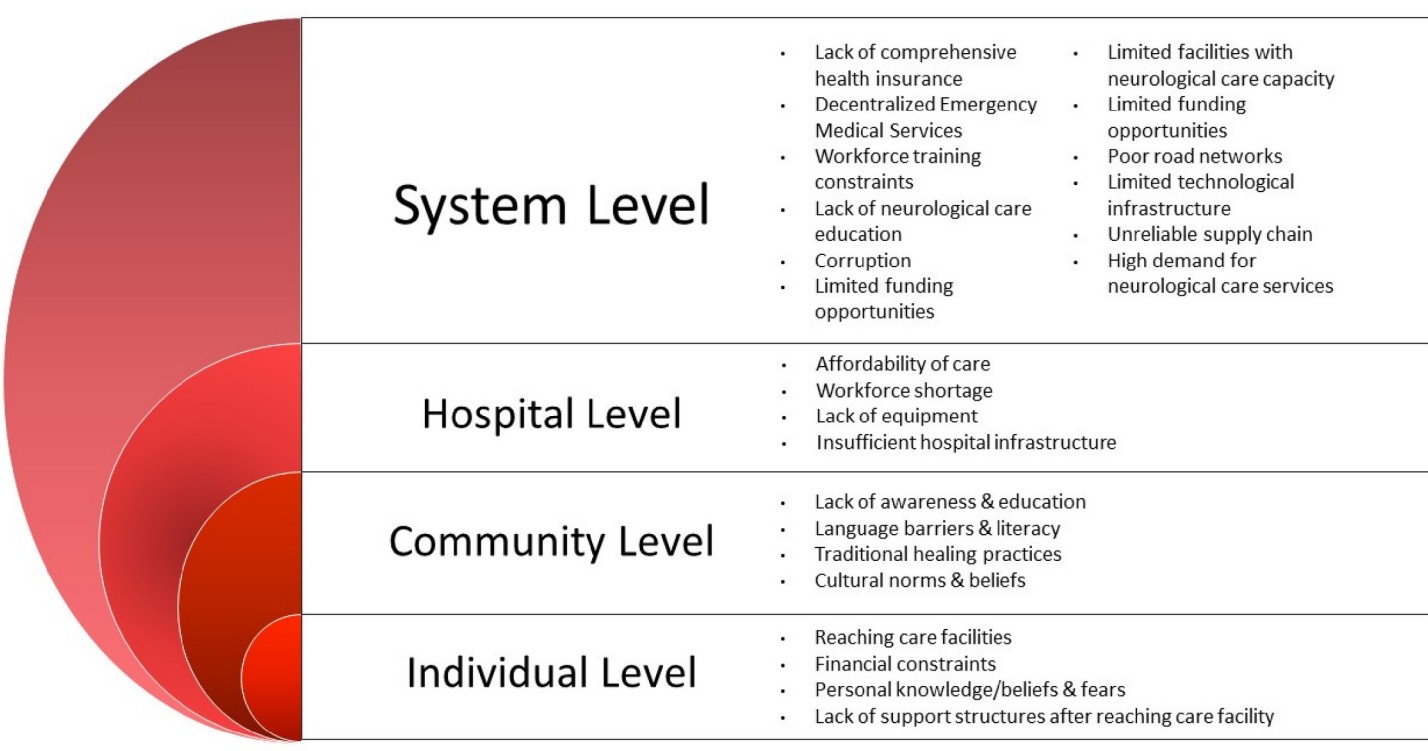

**Fig 5. Multilevel barriers hindering access to neurological care.** Providers described barriers at the individual level, community level, hospital level, and hospital level. These identified barriers collectively prevent neurological patients from initiating, accessing, and receiving necessary neurological care. Such impediments have the potential to harm one's health and incur severe financial burdens.

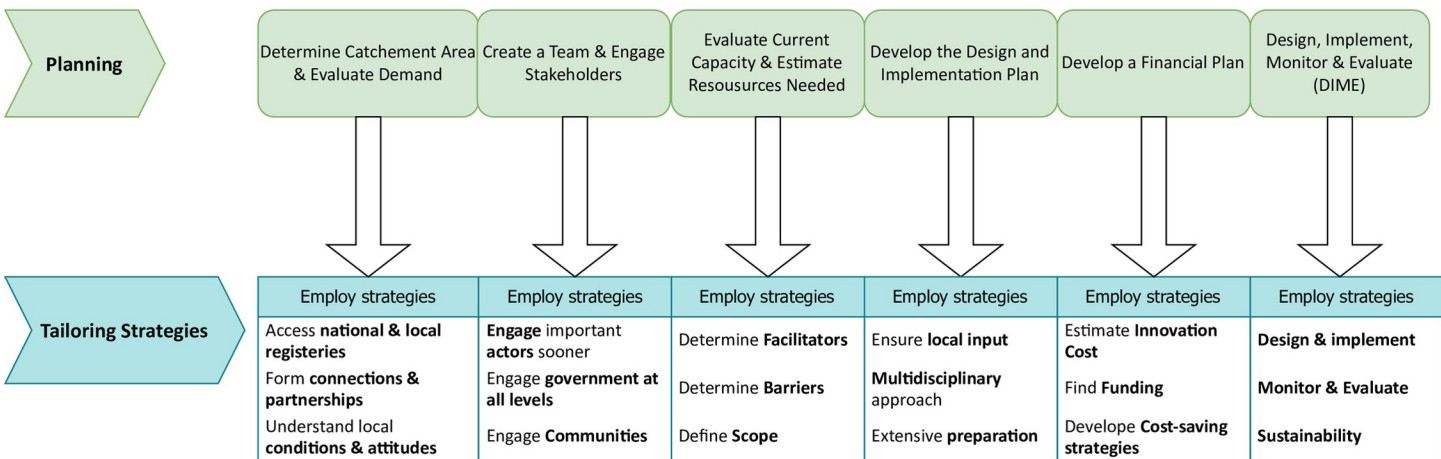

**Fig 6. Strategic planning for successful MNC intervention.** Insights from healthcare providers delve into the realm of planning and tailoring strategies, emphasizing the essential elements required for the successful implementation of MNCs. This diagram outlines key planning considerations that, if carefully executed, have the potential to facilitate widespread adoption by MNCs.

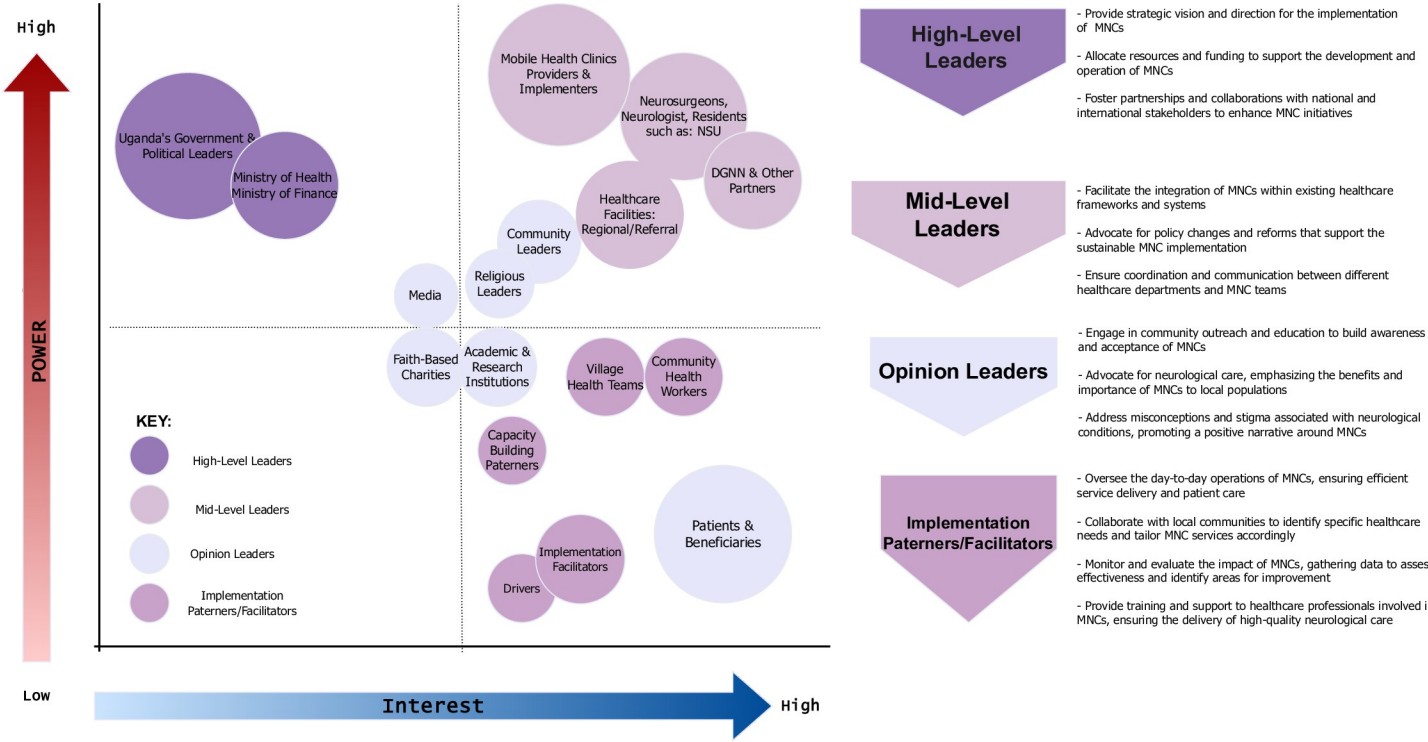

**Fig 7. Stakeholder analysis summarizing actors and their roles.** A power vs. interest analysis of high-level, middle-level, opinion leaders, and implementation facilitators. High-level leaders were identified as decision-makers, executive leaders, or directors with the authority to influence the innovation process. Mid-level leaders, on the other hand, include managers who supervise subordinates and are managed by high-level managers. Their powers are modest, and they play an important role in implementing innovations. Opinion leaders were described as having a subtle influence on the behavior and thinking of others. Religious leaders, local leaders, traditional/cultural leaders, and others are examples of opinion leaders. These actors were described as able to influence the acceptance and implementation of innovations in their communities. Implementation facilitators were identified as on-site experts who provide guidance, coaching, or support during the implementation phase. They include community health workers and medical personnel, who ensure that innovations are effectively implemented and utilized.

*"Involving the Ministry of Health is critical. Mobilizing the community and then identifying these mobile providers, of which there are many: nurses and doctors who want to join and sensitizing them. That, I believe, is significant." (MHC Provider).*

## Discussion

This qualitative study evaluated the feasibility, appropriateness, and usability of mobile neuro clinics (MNCs) to provide neurological care to Uganda's remote and rural populations. Our findings indicate that MNCs were considered feasible, appropriate, and usable for rural and remote communities in Uganda. Respondents emphasized numerous advantages, including demystifying neuroscience; early diagnosis and initiation of neurological care; providing patient follow-up; strengthening referral networks; and, most importantly, improving the country's overall quality of neurological care. In addition to the discerned advantages, healthcare providers also outline strategies for leveraging facilitators while carefully overcoming barriers and engaging stakeholders at every stage of the process.

Our findings further provide additional clarification on the multifaceted substantive design and implementation challenges MNCs could face during the delivery of neurological care. These challenges encompass a spectrum of concerns, barriers, and critical incidents, the origins of which could be attributed to the prevailing context in Uganda. Additionally, providers repeatedly conveyed the intricate nature of neurological care and the absence of explicit policies and regulations governing the activities of MHC providers. However, despite these challenges, MNCs have the potential to improve neurological care by reducing the time, distance, and delay between patients and neurological interventions.

### Feasibility, appropriateness and usability assessed by the sentiment weighted scale

According to the ASS, it was evident that the findings supported the positive valence in all the domains. Consequently, this suggests that MNCs are feasible, appropriate, and usable. Furthermore, the participants identified numerous stakeholders and their roles in the individual domains and characteristics subdomains. The findings suggest that existing facilitators can be leveraged to ensure MNCs' effective design and implementation. **Fig 8** summarizes the concept ideas of the MNC from the providers' suggestions.

### Innovation domain

Most participants were optimistic about the potential benefits that MNCs could provide to the general community, despite several concerns surrounding their adoption. The providers' enthusiastic perspective originates from their previous experiences with mobile health services, recognizing the potential advantages introduced by such advancements. However, the intricate nature of neurological specialties introduces an additional layer of complexity to the design of MNCs. Providers believe that this high level of complexity could hinder the seamless adoption of MNCs in neurological care. As a result, MNC integration necessitates a tailored approach that addresses the complexities specific to neurological care and Uganda. Recognizing and proactively addressing these difficulties is critical to the effective design of MNCs.

Shrime and colleagues (2016) notably advocated developing and deploying mobile surgical units (MSUs) in Uganda [51]. In their research, they explain how MSUs could help Uganda achieve its sustainable development goals (SDGs) and could significantly enhance access to critical surgical procedures. Based on this precedent, MSUs were introduced in Sudan, thereby improving patient access to critical surgical procedures [52]. Furthermore, Qureshi and

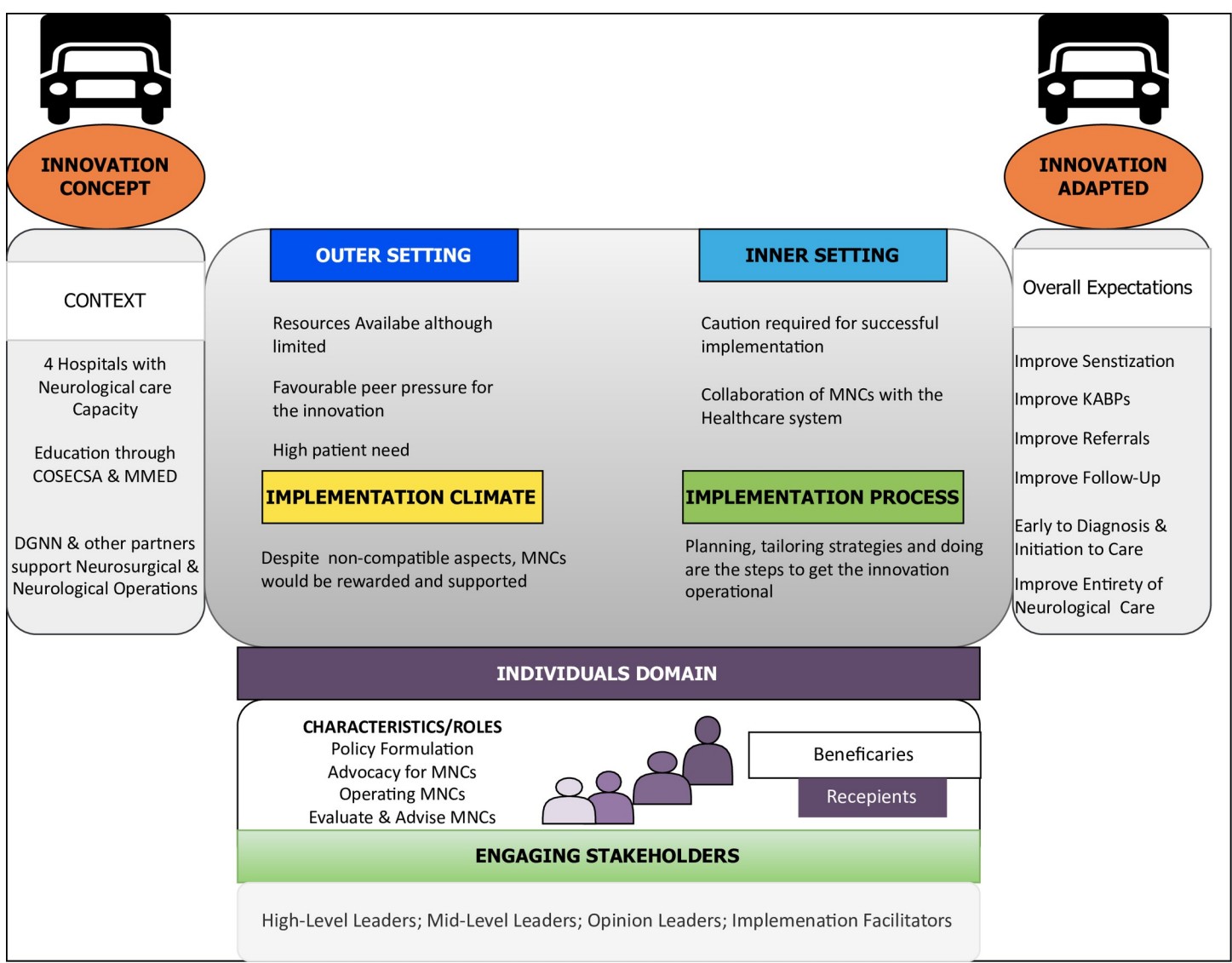

**Fig 8. Schematic illustration depicting a comprehensive conceptual summary emphasizing the major findings of the MNC concept and the approach to adapting based on CFIR.** (left, MNC unadapted; right, Adapted). Findings improve our understanding of the potential impact of MNCs on neurological care. The innovation, when adapted, could meet numerous expectations. It is evident that emphasis must be placed on engaging stakeholders who operate in all domains. "Numerous Challenges exist, but planning, tailoring strategies, and doing are the steps that can get the innovation operational. Evaluating and Reflecting can be the aspects that ensure that the innovation continues to operate efficiently".

colleagues' research signifies the safety, convenience, and cost-effectiveness of mobile endoscopic procedures, providing credibility to MNC's practicality [24, 25, 53]. Similarly, the implementation of mobile clinics in healthcare settings has demonstrated favorable results. For instance, in order to enable follow-up visits and prompt referral of potential clinical issues, BethanyKids, a Christian service provision organization at Kijabe Hospital, set up mobile clinics at 15 locations throughout Kenya [54]. These projects demonstrated the potential of mobile clinics while emphasizing the importance of tailored strategies for the successful development and use of innovative medical treatments. By leveraging the collective expertise accrued from a variety of evidence and real-world applications, there is an opportunity to build high-quality MNCs that effectively address the complexities inherent in the delivery of neurological care.

## Implementation climate constructs

Healthcare providers emphasized the need for change to address the urgent demand for enhanced neurological services while acknowledging that MNCs could fill gaps in the current neurological care system. According to the providers, Uganda's main objective of advancing public health is in line with the proposed MNC concept. Participants acknowledged that MNCs could serve as a long-term factor in the advancement of neurological care. This collective perspective emphasizes MNCs' perceived value and prospective impact, conveying their role as a dynamic and developing part of the larger healthcare framework.

Uganda began its pursuit of neurosurgery in the late 1960s, with groundbreaking work done at Kampala's Mulago Hospital [55]. The practice has grown in scope and spread to CCHU in Mbale, Eastern Uganda and MRRH in Western Uganda [56–58]. Regardless of these establishments, neurological services are still in deficit and are unevenly dispersed. In Uganda's situation, a number of issues include poor access, a high surgical volume, a sparse workforce, and a lack of vital resources in the healthcare system [59, 60]. Addressing this challenge necessitates both innovative curative interventions and preventive measures.

Our study highlights the potential of MNCs as a two-pronged approach to alleviate deficits in neurological care. By delivering essential care directly to communities and emphasizing preventive measures, MNCs have the ability to benefit Ugandans at all levels of society [20, 24, 25, 61]. However, providers expressed multifarious concerns about limited access to neuroscience knowledge, minimal government support, and potential incompatibilities in the components essential for developing and implementing this innovation.

## Outer setting domain

Healthcare providers highlighted a substantial demand for neurological services in Uganda, exacerbated by challenging economic and technological conditions [5, 62]. Additionally, the local attitudes towards neuroscience remain distinctly unfavorable. Given these challenges, providers recognize that the strategic implementation of MNCs could substantially ameliorate the situation. Honoring the importance of collaboration, providers highlighted existing partnerships and underscored the potential for additional cooperation, emphasizing the pivotal role these alliances play in making the vision of MNCs a reality. Nonetheless, participants raised legitimate concerns about potential challenges, with particular emphasis on unforeseen incidents, such as medical errors that could significantly impact the practice of MNCs. Furthermore, the lack of well-established norms and laws for MNCs poses a significant obstacle, highlighting the need for a comprehensive regulatory framework.

Regardless of the uncertainties, global and national initiatives have been launched to improve neurological capacity. Notably, the "4 T's Paradigm" established through Duke and Uganda's collaborative efforts has significantly increased training and technological capacity [55, 63, 64]. The neurological providers expressed gratitude to Duke University, DGNN, CURE International, and numerous other supporters for their ongoing efforts to improve neurosurgery practice. The College of Surgeons of East, Central, and Southern Africa (COSECSA) and the DGNN collaborate to provide specialized neurosurgery training, nurturing a group of practitioners who actively support the nation's neurological care capability [55, 63, 65, 66]. The locally trained Ugandan neurosurgeons could help operate MNCs. In addition, Makerere University, Mbarara University, and some private institutions in Uganda offer the Master of Medicine (MMed) program aimed at improving the neurosurgical workforce [56, 67, 68].

## Inner setting domain

Opinions in the inner setting domain were overwhelmingly positive because providers explained that MNCs that are well-designed and effectively implemented have the potential to integrate seamlessly with the communities and cultures they serve. The findings further emphasized the significance of effective communication and establishing meaningful connections with local populations and healthcare facilities. Successful MNCs, as demonstrated by previous research, [20, 61] hinge on the ability to foster trust and rapport within communities, thus enhancing the acceptance and uptake of neurological services.

Respondents advocated the significance of various structural characteristics in fulfilling the neurological demands of Uganda's rural and remote locations. These include robust physical infrastructure, advanced information technology systems, and a skilled workforce [51, 69]. It is widely recognized that many parts of Uganda face limitations in these structural aspects of health care delivery. MNCs, with their flexibility and adaptability, [70] have the potential to fill these gaps and enhance healthcare support in areas with limited infrastructure in a cost-effective manner. The NeuroPIPES survey corroborates the challenges faced by hospitals across Uganda, indicating shortages in personnel, infrastructure, equipment, and supplies [36]. These shortages directly impact the number of neurosurgery procedures that can be performed, emphasizing the urgent need for innovative solutions such as those from MNCs to bridge these gaps [24, 25, 36, 51, 52, 70].

An informative example pertaining to the impact of MNCs comes from a neurosurgeon with MHC experience. Based on their experience in Ruharo, southwestern Uganda, MHCs deliver service to more than 400 neurosurgery cases annually. Additionally, CCHU in Mbale is successfully operating mobile neurological services in many parts of Uganda [71]. The first-hand testimony and the evidence from practical implementation demonstrate the concrete role that MNCs could play in providing crucial neurological management to marginalized populations.

## Implementation process domain

The landscape of neurological care in Uganda is suffused with a myriad of barriers, spanning individual, community, hospital, and systemic levels [72, 73]. Providers, in particular, described the systemic issues that exist, where a plethora of parties with opposing agendas compete for limited resources [74]. Kapiriri and colleagues (2007) described the contention that Uganda has to consider international interests when setting health priorities, unlike many high-income countries [75]. Consequently, MNCs face stiff competition for resources, similar to other healthcare initiatives in the country, demanding careful strategic planning.

Providers further referenced the intricacy of neurological care as a significant barrier that transcends pure resource allocation challenges. Several researchers have noted that neurological care is multidisciplinary and highly specialized, with unique complexities [76, 77]. This multidisciplinary nature poses challenges in designing and implementing MNCs effectively. The ethical considerations are paramount, given that many neurological conditions necessitate collaboration among multiple healthcare professionals to ensure definitive and quality care [37]. Ford and Kubu (2006) assert that neurological care holds the potential to significantly alter various facets of a patient's personhood, including mood, personality, and cognitive abilities—attributes highly esteemed in patient well-being [78]. As a result, before investing in the design and implementation of MNCs, innovations in neurological care must be carefully planned.

Concerns about the high level of technical specialization in neurological care should also be considered before designing MNCs. Cobb and colleagues (2016) emphasize that quality is

more important than quantity in neurological care. As a result, a high level of technical expertise is required to avoid more than 75% of preventable and technical neurological errors [79]. Therefore, ensuring successful neurological care outcomes requires highly skilled care providers, specialized environments, and equipment. When determining the scope of MNCs, both the interdisciplinary nature and the highly specialized nature of neurological care should be evaluated.

## Stakeholder analysis from individuals domain and sub-characteristics

The study's comprehensive analysis identified four pivotal stakeholder categories in the innovation process: high-level leaders, mid-level leaders, opinion leaders, and implementation facilitators. This comprehensive evaluation elucidates the unique roles and responsibilities inherent to each stakeholder group, providing a solid foundation for a strategic approach designed to ensure optimal design, implementation, and adoption by MNCs. Central to this analysis was the assessment of the stakeholders' varying levels of interest in innovation and health, revealing a spectrum of engagement across the stakeholder landscape.

The Neurosurgical Society of Uganda (NSU), founded in 2017, [80, 81] is a notable example of this. This dynamic organization, comprising prominent neurosurgeons committed to providing high-quality medical care, stands as a beacon of progress in the Ugandan healthcare landscape. The NSU, driven by key neurosurgeons deeply entrenched in the evolution of neurosurgical practices, has taken proactive measures [82]. These dedicated providers have integrated mobile services into their healthcare approaches, recognizing the transformative potential of such initiatives. Their proactive engagement underscores the tangible benefits of adopting mobile services in the country, offering valuable insights into how partnerships can be forged to sustainably support the development and operation of MNCs. This response exemplifies a collaborative approach within the healthcare sector, highlighting the essential role of stakeholder buy-in in driving innovation and fostering sustainability.

Global health actors would also be crucial in influencing the planning and implementation of MNCs in Uganda, in addition to the local stakeholders. Prominent global health organizations, such as the conjoined effort between Duke Global Health Institute (DGHI) and DGNN, [36, 55, 83] Duke University Medical Center, [84] Global Partners in Anesthesia and Surgery (GPAS), [85] Stanford Global Health Neurosurgery, [86] and others, have all shown a dedication to strengthening Uganda's surgical and neurological care infrastructure. To address and raise the caliber of neurological care delivery in the area, a comprehensive strategy should be implemented through the collaboration and active participation of these numerous stakeholders.

## Feasibility, appropriateness, and usability findings

The findings of our study hold significant implications for the successful implementation of MNCs in Uganda. While the innovation concept demonstrates feasibility, appropriateness, and usability, it is essential to acknowledge and address potential concerns and barriers to its effective deployment. Our research underscores the importance of exercising caution throughout the implementation process, emphasizing the need for a meticulous approach to realize the intended benefits. Moreover, the identification of facilitators that can be leveraged to overcome obstacles is crucial. Additionally, the active involvement of stakeholders at all levels is paramount for the sustained success of MNCs in Uganda.

These insights contribute to the development of a comprehensive framework that aligns MNCs with Uganda's unique healthcare landscape, paving the way for improved neurological care delivery. The implications derived from this study reinforce the critical role of strategic

and collaborative approaches in guiding MNCs on their trajectory toward conceptual integration into the Ugandan healthcare system. Strategic planning and collaboration emerge as necessities for the successful integration of MNCs into the existing healthcare system considering the recognized potential and envisaged outcomes.

## Limitations and study strengths

In this study, we employed rigorous inclusion and exclusion criteria for the participants, excluding many healthcare workers, including nurses and anesthesiologists, among others. However, it is important to note that neurological services are multidisciplinary fields that rely on collaboration with various medical specialties and professionals. Hence, gathering insights from other medical specialties involved in neurological care could have added valuable perspectives to the study. Unfortunately, two important healthcare facilities, BNRMH and CCHU, were administratively censored from participating in the study due to logistical and time constraints. BNRMH is known for its expertise in treating epileptic patients, and their inclusion in the study could have provided further enriching insights into neurological care. Similarly, CCHU, a private hospital that incorporates mobile clinic aspects into its services, may have provided greater insight into existing mobile neurological care practices and contributed to a more thorough knowledge of the innovation's design and implementation. Also, the perspectives of other stakeholders, such as policy makers and health sector leaders were not captured, further adding to the limitations of the study.

Additionally, potential bias could have been introduced by basing the calculation of average sentiment scores solely on respondents related to a specific theme or code. As a result, the sentiment analysis might not be representative of the interviewed respondents in a few domains with low response rate from the participants.

Despite these limitations, the study benefited from a robust methodological framework, utilizing the Consolidated Framework for Implementation Research (CFIR). This framework facilitated a comprehensive exploration of key domains and ensured consistency and reliability in data collection. Utilizing the sentiment-weighted scale derived from the CFIR provided a refined dimension to the analysis, ensuring a more precise evaluation of aspects essential to the innovation concept's feasibility, appropriateness, and usability. Moreover, the study maintained a coherent structure with consistent formats for the education and interview sessions, ensuring a seamless transition that enhanced participant understanding and engagement.

A Ugandan expert (B.M.) monitored the study's design and implementation. This method was crucial in maintaining cultural relevance, local context sensitivity, and a comprehensive awareness of the Ugandan healthcare system's complexities. The participation of a Ugandan supervisor provides a dimension of authenticity to the study, improving its quality and contributing to the findings' credibility in the local context.

## Future implications

Our findings provide a suggestive model for designing and implementing MNCs successfully. Based on the suggested model, strategic considerations include completing a needs assessment, engaging stakeholders, and leveraging facilitators. Conducting a detailed needs assessment would provide insight into the community's distinct neurological issues. The strategy will enable MNCs to tailor responses to the specific needs of local populations, thereby increasing the effectiveness and relevance of neurology services. Collaborative efforts at global and national levels will result in a holistic approach that considers different perspectives, resources, and skills, providing a more robust and long-term framework for implementation by MNCs. Finally, by leveraging facilitators, any barriers can be proactively addressed to facilitate the

seamless integration of MNCs into the healthcare landscape while improving the overall quality of neurological care delivery. Future steps include developing a logical conceptual MNC framework, finding funding sources to design and implement MNCs through grants and other sources, and engaging local and global collaborators [87–90].

While MNCs remain feasible, appropriate and usable for enhancing and strengthening neurological care in Uganda, practical alternatives such as telemedicine, community outreach, and awareness campaigns have the ability to supplement MNCs in the short term to strengthen neurological care [20, 91–94]. Long-term planning for neurological care necessitates capacity building at all levels, neurological workforce training, and collaboration with a wide range of stakeholders [55, 63–66] This comprehensive and sustainable healthcare model would provide holistic neurological care while also tailoring it to Uganda's context and challenges.

## Conclusion

In this study, we assessed the feasibility, appropriateness, and usability of MNC's concept in Uganda using the CFIR framework. We discovered that the five CFIR domains demonstrated high positive valence, and the overall ASS was similarly high. Since the average sentiment scores favored positive valence, the MNC's concept was considered feasible, appropriate, and usable. Additionally, the results of the stakeholder analysis examined the power and interests of several stakeholders who ought to be involved in different significant capacities during the establishment and operation of MNCs.

By comprehensively elucidating the barriers and facilitators influencing the feasibility, appropriateness, and usability of Mobile Neuro Clinics (MNCs), as well as delineating the roles of various actors, our study not only provides essential guidelines for the MNC intervention concept but also elucidates the actors who should be engaged throughout the process. Despite several barriers, concerns, and critical incidents, healthcare providers continually emphasized the need for change. They considered the MNC intervention to be aligned with the mission of delivering healthcare to all. As a result, providers developed ways to overcome these limitations to provide quality neurological care through MNCs.

Finally, additional efforts should focus on establishing the specific parameters of MNCs, as well as engaging multiple actors, acknowledging that this endeavor cannot be handled by a single stakeholder. Active participation and transparency should be prioritized throughout the process since they are crucial for designing and putting into practice an innovation that truly benefits all stakeholders. In doing so, stakeholders at the national and global levels will be able to collaborate and develop an innovative MNC.

## Supporting information

**S1 File. Interview guides.**
(DOCX)

**S2 File. Process of data coding and analysis.**
(TIF)

**S3 File. Inclusivity in global health research.**
(DOCX)

**S1 Table. Study constructs.**
(DOCX)

**S2 Table. CodeBook.**
(DOCX)

## Acknowledgments

The authors would like to thank Mulago National Referral Hospital and Mbarara Regional Referral Hospital for being supportive of this research. Special appreciation is extended to Dr. Michael Muhumuza, and Dr. Oscar Obiga for their support with data collection. We would like to appreciate the health care providers who participated in this study. We also thank the DGHI Research Design and Analysis Core (RDAC) for their insightful criticism and assistance during the study's design and analysis stages. We are particularly grateful to Jennifer Headley, Catherine Staton, Joao Ricardo Nickenig Vissoci, and Anna Tuptez. Their perceptive viewpoints and helpful criticisms significantly improved the robustness of our study. We also want to thank the Duke Center for Data and Visualization Sciences. Specifically, Lauren Nichols offered invaluable perspectives into the nuances of our data, which improved the accuracy and depth of our conclusions.

## Author Contributions

**Conceptualization:** Benjamin Mukumbya, Yesel Trillo-Ordonez, Keying Sun, Michael M. Haglund, Anthony T. Fuller.

**Data curation:** Benjamin Mukumbya, Yesel Trillo-Ordonez, Keying Sun, Michael M. Haglund, Anthony T. Fuller.

**Formal analysis:** Benjamin Mukumbya, Yesel Trillo-Ordonez, Keying Sun, Di D. Deng, Kearsley A. Stewart, Michael M. Haglund, Anthony T. Fuller.

**Funding acquisition:** Benjamin Mukumbya, Michael M. Haglund, Anthony T. Fuller.

**Investigation:** Benjamin Mukumbya, Yesel Trillo-Ordonez, Michael M. Haglund, Anthony T. Fuller.

**Methodology:** Benjamin Mukumbya, David Kitya, Yesel Trillo-Ordonez, Keying Sun, Kearsley A. Stewart, Michael M. Haglund, Anthony T. Fuller.

**Project administration:** Benjamin Mukumbya, David Kitya, Michael M. Haglund, Anthony T. Fuller.

**Resources:** Benjamin Mukumbya, David Kitya, Yesel Trillo-Ordonez, Keying Sun, Michael M. Haglund, Anthony T. Fuller.

**Software:** Benjamin Mukumbya, Keying Sun, Michael M. Haglund, Anthony T. Fuller.

**Supervision:** Benjamin Mukumbya, David Kitya, Oscar Obiga, Alvan-Emeka K. Ukachukwu, Michael M. Haglund, Anthony T. Fuller.

**Validation:** Benjamin Mukumbya, David Kitya, Yesel Trillo-Ordonez, Keying Sun, Kearsley A. Stewart, Michael M. Haglund, Anthony T. Fuller.

**Visualization:** Benjamin Mukumbya, Yesel Trillo-Ordonez, Keying Sun, Michael M. Haglund, Anthony T. Fuller.

**Writing – original draft:** Benjamin Mukumbya, David Kitya, Yesel Trillo-Ordonez, Keying Sun, Kearsley A. Stewart, Alvan-Emeka K. Ukachukwu, Michael M. Haglund, Anthony T. Fuller.

**Writing – review & editing:** Benjamin Mukumbya, David Kitya, Yesel Trillo-Ordonez, Keying Sun, Kearsley A. Stewart, Alvan-Emeka K. Ukachukwu, Michael M. Haglund, Anthony T. Fuller.

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
