## [Decision Letter · Decision Letter 0]

20 Feb 2024

PONE-D-23-43695The Feasibility, Appropriateness, and Usability of Mobile Neuro Clinics in Addressing the Neurosurgical and Neurological Demand in UgandaPLOS ONE

Dear Dr. Ukachukwu,

Thank you for submitting your manuscript to PLOS ONE. After careful consideration, we feel that it has merit but does not fully meet PLOS ONE’s publication criteria as it currently stands. Therefore, we invite you to submit a revised version of the manuscript that addresses the points raised during the review process.

We look forward to receiving your revised manuscript.

Kind regards,

Nicholas Aderinto Oluwaseyi

Academic Editor

PLOS ONE

Journal Requirements:

"BM received funding from the Duke Global Health Institute (DGHI), [website: https://globalhealth.duke.edu], and Duke Global Neurosurgery and Neurology (DGNN) [website: https://www.dukeglobalneurosurgery.com]. Members of the DGNN played active roles in the study design, data collection and analysis, manuscript preparation, and decision to publish."

5. We note that [S4 Figure] in your submission contain [map/satellite] images which may be copyrighted. All PLOS content is published under the Creative Commons Attribution License (CC BY 4.0), which means that the manuscript, images, and Supporting Information files will be freely available online, and any third party is permitted to access, download, copy, distribute, and use these materials in any way, even commercially, with proper attribution. For these reasons, we cannot publish previously copyrighted maps or satellite images created using proprietary data, such as Google software (Google Maps, Street View, and Earth). For more information, see our copyright guidelines: http://journals.plos.org/plosone/s/licenses-and-copyright.

a. You may seek permission from the original copyright holder of S4 Figure to publish the content specifically under the CC BY 4.0 license.  

Reviewers' comments:

Reviewer's Responses to Questions

**Comments to the Author**

1. Is the manuscript technically sound, and do the data support the conclusions?

Reviewer #1: Yes

Reviewer #2: Yes

2. Has the statistical analysis been performed appropriately and rigorously? 

Reviewer #1: Yes

Reviewer #2: N/A

3. Have the authors made all data underlying the findings in their manuscript fully available?

Reviewer #1: No

Reviewer #2: No

4. Is the manuscript presented in an intelligible fashion and written in standard English?

Reviewer #1: Yes

Reviewer #2: No

5. Review Comments to the Author

Reviewer #1: Thank you for the opportunity to review this article on examining the feasibility, appropriateness, and usability of mobile neuro clinics (MNCs) in Uganda. I found the article to be very well written, and I especially appreciated the clearly described Ugandan context at multiple levels, which would help readers less familiar with the context better understand the various influences at play when implementing an intervention such as MNCs. The authors provide a good reason for not making available all data underlying their findings. My suggestions below are mostly to provide additional explanations for the reader regarding the use of the sentiment-weighted scale and further considerations in moving the MNC effort forward.

1. Please consider explaining how it was decided that the positive valence of the sentiment-weighted scale signals feasibility, appropriateness, and usability of MNCs. Each of these measures have several components that are often considered to be part of the whole measure – e.g.: usability is commonly thought to encompass relative efficiency, ease of learning, etc.; and feasibility is commonly thought to encompass perceived burden, reasons for non-use, etc. It would be helpful to see a clearer articulation of how the data and approaches to analysis enabled assessment of such multiple components per measure to deem MNCs feasible, appropriate, and usable.

2. Please consider addressing potential biases introduced from basing the calculation of average sentiment scores only on those who responded to a particular theme/code.

3. Please consider elaborating in the Discussion section on practical alternatives to MNCs in addressing Uganda’s neurosurgical/neurological care needs, as such alternatives’ relative advantages and disadvantages can be expected to be a key consideration in preparing for MNC implementation.

4. Please consider including as a limitation how the study data do not speak to perspectives of non-frontline providers (e.g., administrators/leadership at multiple levels), similar to how the article already acknowledges how the study data do not include insights from other medical specialties.

5. (Minor) Second paragraph of the Data Collection section: The 1st sentence seems to say that the first section of the interviews was about demographic information, while the 2nd sentence seems to say that the first section assessed providers’ understanding of healthcare. Please consider clarifying.

6. (Minor) Last paragraph of the Introduction section, 2nd sentence: Please consider checking this sentence for any unintended grammatical errors.

Reviewer #2: Overall

Thank you for inviting me to review manuscript ID PONE-D-23-43695. This manuscript describes the potential of mobile health clinics as a strategy for improving access to neurological services for people in rural and remote areas of Uganda. I congratulate the authors on conducting an important study. In my opinion, the manuscript requires major revisions prior to being considered suitable for publication. I hope my reflections and suggestions help to strengthen the manuscript.

Abstract

The abstract is well written.

Introduction

The section introduces the concept of inequitable access to healthcare for people living in rural and remote communities. However, I wonder if the authors would consider teasing out the specific barriers or complexities surrounding access to healthcare in these communities, as noted in the first paragraph. The authors draw on mobile health clinic literature from different countries to argue the need for the adaption of MNCs in Uganda. The sentence straddling pages 3 and 4 may require revisiting to ensure clarity. The sentence following uses an acronym (CCHU) without first writing the term out in full.

Methods

The methods section is written clearly. It surprises me that ethics exemption was provided, given the study captured the experiences of health professionals delivering services in rural areas, where it is common for the anonymity of rural people to be reduced via dual relationships. The first sentence in the ethics statement section on page 5 requires revisiting for clarity.

Would the authors consider adding information about the primary geographic work location of participants in terms of rurality? In Australia, MMM categories are used to categorize rurality (https://www.health.gov.au/topics/rural-health-workforce/classifications/mmm), although there may be a more suitable approach to use.

On page 6, the authors note that thematic analysis was conducted to develop the code books, however it is not clear how many transcripts were used to identify codes and/or themes comprised in the code books.

The last sentence under the title “Figure 1” on page 7 requires some attention for clarity.

Results

The results section, along with the figures, describes the data according to the CFIR domains and stakeholder analysis. The figures capture significant detail that could be teased out into paragraphs for the reader.

It might also be useful to include more detail within the stakeholder analysis section, if possible.

Discussion

The authors have broken the discussion section into individual sections relating to the CFIR domains and the sentiment weighted scale. Under many of these headings, the authors have provided interesting content that, in my perspective, fit better under the respective headings in the results section (the first discussion paragraph under the heading ‘Innovation Domain’ starting on page 17 is a good example of this). The authors may wish to consider reworking their discussion section by removing subheadings, revisiting their study aim on page 4, and using this section along with the results to identify and address the main discussion points arising from the study. This will help to give the section direction and clarity.

Conclusion

The first sentence in the conclusion does not mention the topic of the study, but instead, notes the valence of the CFIR domains. The authors may wish to revise this.

6. PLOS authors have the option to publish the peer review history of their article (what does this mean?). If published, this will include your full peer review and any attached files.

Reviewer #1: **Yes: **Bo Kim

Reviewer #2: No

---

## [Author Response · Author response to Decision Letter 0]

25 Apr 2024

Title of the Manuscript: The Feasibility, Appropriateness, and Usability of Mobile Neuro Clinics in Addressing the

Neurosurgical and Neurological Demand in Uganda

Manuscript Number: [PONE-D-23-43695]

Reviewer 1

Comment 1 Reviewer #1: Thank you for the opportunity to review this article on examining the feasibility, appropriateness, and usability of mobile neuro clinics (MNCs) in Uganda. I found the article to be very well written, and I especially appreciated the clearly described Ugandan context at multiple levels, which would help readers less familiar with the context better understand the various influences at play when implementing an intervention such as MNCs. The authors provide a good reason for not making available all data underlying their findings. My suggestions below are mostly to provide additional explanations for the reader regarding the use of the sentiment-weighted scale and further considerations in moving the MNC effort forward.

Authors’ Response: Thank you for the wonderful observations and comments regarding our manuscript. 

Change to Text: N/A

Comment 2: 1. Please consider explaining how it was decided that the positive valence of the sentiment-

weighted scale signals feasibility, appropriateness, and usability of MNCs. Each of these measures have several components that are often considered to be part of the whole measure – e.g.: usability is commonly thought to encompass relative efficiency, ease of learning, etc.; and feasibility is commonly thought to encompass perceived burden, reasons for non-use, etc. It would be helpful to see a clearer articulation of how the data and approaches to analysis enabled assessment of such multiple components per measure to deem MNCs feasible, appropriate, and usable.

Authors’ Response: Thank you for this insightful observation. Adopting CFIR as a guiding framework enhances the rigor and applicability of implementation research, facilitating the translation of scientific discoveries into effective real-world practices. We have included a clarifying statement in the Methods section to buttress this point.

Change to Text: “As such, the CFIR framework enables a holistic examination of the complex interplay of factors influencing MNCs and their successful implementation. Furthermore, the study included healthcare providers with varying experience levels, the CFIR framework was ideal for thoroughly evaluating the participants' perspectives.”

Comment 3: 2. Please consider addressing potential biases introduced from basing the calculation of average sentiment scores only on those who responded to a particular theme/code.

Authors’ Response: Thank you for this suggestion. We agree that basing the calculation on average sentiment scores may introduce potential biases and have included additional details in the limitation section.

Change to Text: “Additionally, potential bias could have been introduced by basing the calculation of average sentiment scores solely on respondents related to a specific theme or code. As a result, the sentiment analysis might not be representative of the interviewed respondents in a few domains with low response rate from the participants.”

Comment 4: 3. Please consider elaborating in the Discussion section on practical alternatives to MNCs in addressing Uganda’s neurosurgical/neurological care needs, as such alternatives’ relative advantages and disadvantages can be expected to be a key consideration in preparing for MNC implementation.

Authors’ Response: Thank you for this observation. We have added a detailed explanation on how MNCs can be integrated into the health system.

Change to Text: “While MNCs remain feasible, appropriate, and usable for enhancing and strengthening neurological care in Uganda, practical alternatives such as telemedicine, community outreach, and awareness campaigns have the ability to supplement MNCs in the short term to strengthen neurological care. Long-term planning for neurological care necessitates capacity building at all levels, neurological workforce training, and collaboration with a wide range of stakeholders. This comprehensive and sustainable healthcare model would provide holistic neurological care while also tailoring it to Uganda's context and challenges.”

Comment 4: Please consider including as a limitation how the study data do not speak to perspectives of

non-frontline providers (e.g., administrators/leadership at multiple levels), similar to how the article already acknowledges how the study data do not include insights from other medical specialties.

Authors’ Response: Thank you for this suggestion. We agree that this is vital for the study. We have administrators for MHCs and if you look at the stakeholder analysis, we further provide details on many other stakeholders. In the future, we also intend to engage policymakers as the project takes off. (page 4-5: Setting and Participants). We have included a statement in the Limitations section noting this excluded demographic.

Change to Text: “Also, the perspectives of non-frontline providers, such as administrators and health leaders were not captured, further adding to the limitations of the study.”

Comment 5: (Minor) Second paragraph of the Data Collection section: The 1st sentence seems to say that the first section of the interviews was about demographic information, while the 2nd sentence seems to say that the first section assessed providers’ understanding of healthcare. Please consider clarifying.

Authors’ Response: Thank you for this observation. We have corrected it accordingly.

Change to Text: “We conducted in-depth face-to-face interviews in which participants answered open-ended questions before being probed to clarify any additional information. Interview guides collected demographic information, and also had four different sections. The four sections collected information related to the understanding of healthcare, MHCs and their application in neurological care and other general medical specialties, and the barriers and facilitators to designing and implementing MNCs. The first part of the interview guide for MHC providers assessed their general understanding of neurosurgery and neurology, while that of neurological specialists assessed their general understanding of mobile clinics. The remaining three parts of both interview guides followed the same structure.”

Comment 6: 6. (Minor) Last paragraph of the Introduction section, 2nd sentence: Please consider checking

this sentence for any unintended grammatical errors.

Authors’ Response: Thank you for this observation. We have corrected the statement accordingly.

Change to Text: “A pediatric neurosurgical MHC successfully designed and implemented by Owler and colleagues in rural and remote Australia improved neurosurgical outcomes for patients by decreasing the delays to definitive neurological care.”

Reviewer 2

Comment 1: Thank you for inviting me to review manuscript ID PONE-D-23-43695. This manuscript

describes the potential of mobile health clinics as a strategy for improving access to neurological services for people in rural and remote areas of Uganda. I congratulate the authors on conducting an important study. In my opinion, the manuscript requires major revisions prior to being considered suitable for publication. I hope my reflections and suggestions help to strengthen the manuscript.

Abstract

The abstract is well written.

Authors’ Response: Thank you for this kind observation and commendation of our manuscript.

Change to Text: N/A

Comment 2a: Introduction

The section introduces the concept of inequitable access to healthcare for people living in rural and remote communities. However, I wonder if the authors would consider teasing out the specific barriers or complexities surrounding access to healthcare in these communities, as noted in the first paragraph. The authors draw on mobile health clinic literature from different countries to argue the need for the adaption of MNCs in Uganda. The sentence straddling pages 3 and 4 may require revisiting to ensure clarity. The sentence following uses an acronym (CCHU) without first writing the term out in full.

Authors’ Response: Thank you for this suggestion. We believe the paper describes these aspects in further detail towards the end of the first paragraph of the introduction section. Here is an example highlighting these barriers: “Given that Uganda only has four neurological hospitals, providing neurological care across the country is impossible. Uganda's demand for specialized neurological services exceeds its supply due to inadequate infrastructure and workforce.”

We further add two sentences to emphasize the barriers in our study.

Change to Text: The aforementioned barriers lead to disparities in the quality of neurological care for people in rural and remote communities, signifying an urgent need for innovative approaches to neurological service delivery. These approaches should transcend geographical isolation and socioeconomic factors to address challenges such as limited access to neurological care, shortage of trained healthcare professionals, and inadequate infrastructure.

Comment 2b: Introduction

The sentence following uses an acronym (CCHU) without first writing the term out in full.

Authors’ Response: Thank you for this suggestion. We have written out the acronym, CCHU, in full as advised.

Change to Text: “CURE Children’s Hospital of Uganda (CCHU)”

Comment 3: Methods

The methods section is written clearly. It surprises me that ethics exemption was provided, given the study captured the experiences of health professionals delivering services in rural areas, where it is common for the anonymity of rural people to be reduced via dual relationships. The first sentence in the ethics statement section on page 5 requires revisiting for clarity.

Authors’ Response: Thank you for this comment. We obtained ethical approval from both Duke Health IRB and Mulago Hospital REC after review of the study protocol, instruments, and consent forms.

Change to Text: “The study received ethical approval from the Duke Health Institutional Review Board (IRB #: Pro00110750) and from the Mulago Hospital Research Ethics Committee (MHREC 2317).”

Comment 4a: Would the authors consider adding information about the primary geographic work location of

participants in terms of rurality? In Australia, MMM categories are used to categorize rurality https://www.health.gov.au/topics/rural-health-workforce/classifications/mmm), although there may be a more suitable approach to use. 

Authors’ Response: Thank you for sharing the MMM report, we do not think that it fits with this current study, but it is definitely a concept we will adopt as we start designing and implementing MNCs. Moreover, identifying the primary geographic work location of our participants would negate our decision to keep their identifying details confidential, given our small sample size. In regard to the comment on thematic analysis, we have revised it accordingly. Additionally, thank you for catching that error on page 7 regarding the Figure 1 caption. The confusing aspect has been deleted since it might have been an error during editing.

Change to Text: N/A

Comment 4b: On page 6, the authors note that thematic analysis was conducted to develop the code books,

however it is not clear how many transcripts were used to identify codes and/or themes comprised in the code books.

The last sentence under the title “Figure 1” on page 7 requires some attention for clarity.

Authors’ Response: Regarding the comment on thematic analysis, we have revised it accordingly. Additionally, thank you for catching that error on page 7 regarding the Figure 1 caption. The confusing aspect has been deleted since it might have been an error during editing.

Change to Text: “We created two distinct codebooks from 5 sampled neurological and 4 sampled MHC providers’ interviews, respectively. (Refer to S3 Process of Data Coding and Analysis for further breakdown)”

Comment 5: Results

The results section, along with the figures, describes the data according to the CFIR domains and stakeholder analysis. The figures capture significant detail that could be teased out into paragraphs for the reader. It might also be useful to include more detail within the stakeholder analysis section, if possible.

Authors’ Response: Thank you for this wonderful observation and suggestion. We believe the Figure 7: Stakeholder Analysis has a detailed explanation of all the described stakeholders. Rewriting it in the caption would not offer any merit since we would describe the same details as the ones described in the figure.

Change to Text: N/A

Comment 6: Discussion

The authors have broken the discussion section into individual sections relating to the CFIR domains and the sentiment weighted scale. Under many of these headings, the authors have provided interesting content that, in my perspective, fit better under the respective headings in the results section (the first discussion paragraph under the heading ‘Innovation Domain’ starting on page 17 is a good example of this). The authors may wish to consider reworking their discussion section by removing subheadings, revisiting their study aim on page 4, and using this section along with the results to identify and address the main discussion points arising from the study. This will help to give the section direction and clarity.

Authors’ Response: Thank you for this observation and suggestion. We revised the only subheading that needed clarification to meet this demand. Additionally, we believe that the subheadings are still helpful in guiding the readers since the paper has many moving parts that can be overwhelming without the subheadings.

Change to Text: “Feasibility, Appropriateness and Usability Assessed by the Sentiment Weighted Scale”

Comment 7: Conclusion

The first sentence in the conclusion does not mention the topic of the study, but instead, notes the valence of the CFIR domains. The authors may wish to revise this.

Authors’ Response: Thank you for this observation. The order of the sentences has been changed so that it is clear.

Change to Text: “In this study, we assessed the feasibility, appropriateness, and usability of MNC’s concept in Uganda using the CFIR framework. We discovered that the five CFIR domains demonstrated high positive valence, and the overall ASS was similarly high. Since the average sentiment scores favored positive valence, the MNC's concept was considered feasible, appropriate, and usable.”

Editor-in-Chief

Comment 1: Thank you for stating the following financial disclosure: "BM received funding from the Duke Global Health Institute (DGHI), [website: https://globalhealth.duke.edu], and Duke Global Neurosurgery and Neurology (DGNN) [website: https://www.dukeglobalneurosurgery.com]. Members of the DGNN played active roles in the

study design, data collection and analysis, manuscript preparation, and decision to publish."

Authors’ Response: Thank you for pointing this dilemma out. We have fixed the funding section to clarify that funding was from Duke Global Health Institute (DGHI), [website: https://globalhealth.duke.edu] and the details are in the cover letter.

Change to Text: Funding:

This study was supported by the Duke Global Health Institute.

Comment 2: Please provide a complete Data Availability Statement in the submission form, ensuring you include all necessary access information or a reason for why you are unable to make your data freely accessible. If your research concerns only data provided within your submission, please write "All data are in the manuscript and/or supporting information files" as your Data Availability Statement.

Author’s Response: Thank you for this observation. We have added a Data Availability Statement that will be submitted with our resubmission.

Change to Text: “We have included all relevant information for our findings but cannot make all data readily accessible due to the risk of identifying persons or organizations.”

Comment 3: We note that [S4 Figure] in your submission contain [map/satellite] images which may be copyrighted. All PLOS content

---

## [Decision Letter · Decision Letter 1]

30 May 2024

The Feasibility, Appropriateness, and Usability of Mobile Neuro Clinics in Addressing the Neurosurgical and Neurological Demand in Uganda

PONE-D-23-43695R1

Dear Dr. Ukachukwu,

We’re pleased to inform you that your manuscript has been judged scientifically suitable for publication and will be formally accepted for publication once it meets all outstanding technical requirements.

Kind regards,

Nicholas Aderinto Oluwaseyi

Academic Editor

PLOS ONE

Additional Editor Comments (optional):

Reviewers' comments:

Reviewer's Responses to Questions

**Comments to the Author**

1. If the authors have adequately addressed your comments raised in a previous round of review and you feel that this manuscript is now acceptable for publication, you may indicate that here to bypass the “Comments to the Author” section, enter your conflict of interest statement in the “Confidential to Editor” section, and submit your "Accept" recommendation.

Reviewer #1: All comments have been addressed

2. Is the manuscript technically sound, and do the data support the conclusions?

Reviewer #1: Yes

3. Has the statistical analysis been performed appropriately and rigorously? 

Reviewer #1: Yes

4. Have the authors made all data underlying the findings in their manuscript fully available?

Reviewer #1: No

5. Is the manuscript presented in an intelligible fashion and written in standard English?

Reviewer #1: Yes

6. Review Comments to the Author

Reviewer #1: (No Response)

7. PLOS authors have the option to publish the peer review history of their article (what does this mean?). If published, this will include your full peer review and any attached files.

Reviewer #1: **Yes: **Bo Kim

---

## [Editor Report · Acceptance letter]

13 Jun 2024

PONE-D-23-43695R1 

PLOS ONE

Dear Dr. Ukachukwu, 

I'm pleased to inform you that your manuscript has been deemed suitable for publication in PLOS ONE. Congratulations! Your manuscript is now being handed over to our production team.

Kind regards, 

on behalf of

Dr. Nicholas Aderinto Oluwaseyi 

Academic Editor

PLOS ONE